# Variants Identified in the *HOXC13* and *HOXD13* Genes Suggest Association with Cervical Cancer in a Cohort of Mexican Women

**DOI:** 10.3390/genes14020358

**Published:** 2023-01-30

**Authors:** Karina Janett Juárez-Rendón, Manuel Alejandro Castro-García, Diddier Giovanni Prada-Ortega, Gildardo Rivera, Luz María Ruíz-Godoy, Virginia Isabel Enríquez-Cárcamo, Miguel Angel Reyes-Lopez

**Affiliations:** 1Centro de Biotecnología Genómica, Instituto Politécnico Nacional, Blvd. del Maestro s/n. Esq. Elías Piña. Col. Narciso Mendoza, Reynosa 88710, Mexico; 2Department of Environmental Health Sciences, Columbia University Mailman School of Public Health, New York, NY 10027, USA; 3Unit for Biomedical Research in Cancer, Instituto Nacional de Cancerología, México City 14080, Mexico; 4Department of Biomedical Informatics, Faculty of Medicine, Universidad Nacional Autónoma de México, México City 04510, Mexico; 5Banco de Tumores, Instituto Nacional de Cancerología, México City 14080, Mexico

**Keywords:** cervical cancer, *HOXC13*, *HOXD13*, gene variants, Mexican population and bioinformatics

## Abstract

*HOX* genes have been associated with carcinogenesis. However, the molecular mechanism by which tumors are generated remains unclear. The *HOXC13* and *HOXD13* genes are of interest for their involvement in the development of genitourinary structures. The aim of this first study in the Mexican population was to search for and analyze variants in the coding region of the *HOXC13* and *HOXD13* genes in women with cervical cancer. Samples from Mexican women with cervical cancer and healthy women were sequenced (50/50). Allelic and genotypic frequencies were compared between groups. The functional impact of the proteins was determined with two bioinformatics servers (SIFT and PolyPhen-2), and the oncogenic potential of the identified nonsynonymous variants was determined using the CGI server. We identified five unreported gene variants: c.895C>A p.(Leu299Ile) and c.777C>T p.(Arg259Arg) in the *HOXC13* gene and c.128T>A p.(Phe43Tyr), c.204G>A p.(Ala68Ala), and c.267G>A p.(Ser89Ser) in the *HOXD13* gene. In this study, we suggest that the non-synonymous variants c.895C>A p.(Leu299Ile) and c.128T>A p.(Phe43Tyr) could represent a risk factor for the development of the disease, although additional studies in larger patient populations and in different ethnic groups are needed in order to support the results observed.

## 1. Introduction

Cervical cancer (CC) is a gynecological condition of multifactorial origin, characterized by abnormal growth of cells in the cellular lining of the cervix, although precancerous lesions usually begin in the transformation zone, where the endocervix and exocervix are connected [1]. Considering the histopathological classification of CC, squamous cell carcinoma is the most frequent subtype (75%), while adenocarcinomas are observed in fewer patients (5–20%) [2]. With an estimated incidence of 570,000 cases and 311,000 deaths in 2018 alone, CC has been reported as the fourth most common type of cancer in women worldwide, making this disease a global health problem [3]. In some countries, the number of cases has been decreasing thanks to prevention, detection, and timely treatment programs. However, countries with low socioeconomic status and restricted access to medical care show greater morbidity and mortality [4]. In Mexico, CC ranked third in incidence in 2018 (11 cases per 100,000 women) and second in mortality (5.8 cases per 100,000), thus having a prevalence of 22,000 cases in the last 5 years [5].

CC is a complex disease that has been strongly associated with recurrent human papillomavirus (HPV) infections (95% of cases), especially with high-risk types (e.g., 16 and 18). However, there is a small clinical group of HPV-negative cases [6,7], suggesting that HPV infection may be necessary but not sufficient to cause malignant transformation in cervical cells [8]; therefore, the co-participation of several factors (e.g., environmental, epigenetic and/or genetic factors) could be involved in its development [9,10]. Smoking, for example, is considered a risk factor for the development of the disease, mainly because tobacco by-products may alter the structure of DNA in cervical cells. Thus, women who smoke have twice the risk of developing CC compared to non-smokers [11]. DNA methylation or histone modification are also triggers of the disease [10]. However, the genetic factor (as a non-modifiable element) is of particular interest in research, especially because it has been associated with accumulated DNA damage favoring the deregulation or abnormal expression of oncogenes or tumor suppressor genes, which play important roles in cell-cycle progression, chromosomal stability, cell proliferation and differentiation, immune responses, and apoptosis [3,8]. On the other hand, previous studies have observed familial aggregation in patients with CC, determining that the risk of developing the disease is twice as high in first-degree relatives compared to other relatives. In addition, it has been established that genetic heritability is higher in CC (27%) than in other neoplasms [12], and also because some variants (mutations/polymorphisms) identified in candidate genes could potentially contribute to the genetic susceptibility of CC among individuals [13].

The homeobox (*HOX*) gene family encodes highly conserved transcription factors, which were first identified in *Drosophila melanogaster* as the *HOX*-C complex and subsequently discovered in other species, including mammals [14]. *HOX* genes play a crucial role in embryonic and adult development, regulating cell differentiation, proliferation, angiogenesis, motility, and apoptosis [15]. However, the altered expression of these genes has been associated with the onset and development of various types of cancer such as lung, ovarian, prostate, cervix, breast, neuroblastoma, head and neck cancer, and leukemia [16,17]. Thus, its involvement in diagnosis and treatment should be considered [15].

Structurally, *HOX* genes are simple. They consist of two exons and a single intron. In the second exon is the homeodomain, a 120-nucleotide region coding for the 61-amino-acid *HOX* protein, characterized by its high homology in the paralogous groups [18,19]. In humans, the *HOX* genes are organized into classes. Class I consists of 39 genes, divided into four paralogous groups, A, B, C, and D, located at loci *HOXA*: 7p15.3, *HOXB*: 17q21.3, *HOXC*: 12q13.3, and *HOXD*: 2q31-32. Each group contains 9 to 11 genes that have been numbered from 1 to 13 (non-consecutively) based on both their sequence and position within the group [18,20]. The expression of these genes is spatiotemporal. The 3′ genes are expressed in the anterior region, while the 5′ genes are expressed in the posterior region [21]. The latter genes are involved in the differentiation and development of the digestive tract and the genitourinary region [20]. Particularly, the *HOXC13* and *HOXD13* genes belong to this group. Their aberrant expression and their association with various types of cancer have made them candidate genes for the study of neoplasms [22].

The *HOXC13* and *HOXD13* genes encode for proteins (with the same name) of 330 and 343 amino acids, respectively. In both genes, exon 1 encodes the first 757 bp, of which 45 bp corresponds to a region of 15 polyalanines. Exon 2 encodes 250 bp (758 to 1008), although 180 bp belongs to the homeodomain [23]. Quantitative real-time polymerase chain reaction (qRT-PCR) and Western blot assays have demonstrated that *HOXC13* is highly expressed in cervical cancer [24], metastatic melanoma, liposarcoma, glioblastoma, sarcomas, and esophageal, skin, prostate [25,26], and breast cancers [27]. In lung cancer, *HOXC13* has been associated with severe clinical features and poor prognosis; therefore, this gene has been proposed as a strong oncogene candidate [28]. On the other hand, the role of *HOXD13* is still controversial. The dysregulation of this gene and tumor formation have been observed in melanoma, astrocytoma, and cervical and breast cancers. The latter has been associated with a poor prognosis [29]. However, in prostate cancer, *HOXD13* acts as a tumor suppressor gene [30]. Considering the participation of these genes in the development of genitourinary structures and their background in carcinogenesis, this study sought and analyzed *HOXC13* and *HOXD13* gene variants in Mexican women with CC.

## 2. Materials and Methods

### 2.1. Samples

A total of 100 CC and healthy samples (50/50) were analyzed from Mexican women older than 18 years of age, all of whom provided their informed consent (002/2017/CEI). The patient samples were provided by the Tumor Bank of the Instituto Nacional de Cancerología (INCan) in Mexico City, and the healthy population samples were taken from volunteers of diverse origins within the country.

The patient group included women with CC at any clinical stage, without comorbidities, and with a positive or negative history of cancer. The procedures hereby described were performed following the guidelines and ethical standards established in the Declaration of Helsinki. The study was descriptive, observational, and comparative.

### 2.2. Molecular Analysis

Biopsy and peripheral blood were obtained from each affected participant, while in healthy women only the latter was analyzed. Genomic DNA was extracted using a standard protocol (Miller et al., 1988) [31]. The coding region of the *HOXC13* and *HOXD13* genes (NCBI, ID:3229 and 3239, respectively), divided into fragments 1, 2A, and 2B, was amplified by polymerase chain reaction (PCR). The primers were designed de novo (Table 1) and verified in silico using the University of California Santa Cruz (UCSC) server (https://genome.ucsc.edu/cgi-bin/hgPcr, accessed on 29 December 2022). The amplicons were analyzed by 1% agarose gel electrophoresis and sequenced in triplicate using the Sanger method (ABI 3130; Applied Biosystems, Foster City, CA, USA).

### 2.3. Statistical Analysis

Descriptive statistics (means and frequencies) were used in the analysis of clinical-demographic variables. The observed genotypes were counted directly, and the allelic and genotypic frequencies were analyzed with the Arlequin software v3.0, comparing the groups (patients vs. healthy women) with a χ2 test (*p* ≤ 0.05) (SPSS v23, Inc., Chicago, IL, USA).

### 2.4. Bioinformatics Analysis

The wild-type sequences of the *HOXC13* and *HOXD13* genes were obtained from the UCSC server and then aligned with the gene sequence from each patient using the Molecular Evolutionary Genetics Analysis (MEGA) software (v.11.0) in order to confirm the nucleotide change position. The bioinformatics servers Sorting Intolerant From Tolerant (SIFT) and Polymorphism Phenotyping-2 (PolyPhen-2), available at: https://sift.bii.a-star.edu.sg and http://genetics.bwh.harvard.edu/pph2/, accessed on 29 December 2022) respectively, were used to predict the impact of any amino acid changes on protein function (non-synonymous variants). Additionally, the Cancer Genome Interpreter (CGI) server (https://www.cancergenomeinterpreter.org/analysis accessed on 29 December 2022) was used to determine their possible oncogenic potential.

## 3. Results

### 3.1. Molecular and Bioinformatics Analysis

We identified five unreported gene variants in the coding region of the *HOXC13* (*n* = 2) and *HOXD13* (*n* = 3) genes (Figure 1). The non-synonymous transversion c.895C>A p.(Leu299Ile) and the synonymous transition c.777C>T p.(Arg259Arg) were found in the *HOXC13* gene, each observed in a heterozygous state in one patient (2%). Both were found within exon 2A and were not present in the healthy women’s group. On the other hand, the non-synonymous transversion c.128T>A p.(Phe43Tyr) was observed in exon 1 of the *HOXD13* gene. This gene variant was observed in a heterozygous state in 11 patients (22%) and was absent in the healthy women’s group. The synonymous transitions c.204G>A p.(Ala68Ala) and c.267G>A p.(Ser89Ser) were also present in the patients (observed in exon 1, both in heterozygous state) and in the healthy female group (homozygous and heterozygous state). The allelic and genotypic frequencies comparison between groups was significant (*p* = 0.001) for the c.128T>A variant only (Table 2). The bioinformatics analysis demonstrated that the c.895C>A p.(Leu299Ile) and c.128T>A p.(Phe43Tyr) transversions have a deleterious effect on protein function, with scores of 0 (Damaging) and 1 (Probably Damaging) for the former (SIFT and PolyPhen-2, respectively) and 0.002 (Damaging) and 0.898 (Possibly Damaging) for the latter (Table 3). The two variants were classified by the CGI server as Passenger-type mutations. No gene variants were identified in exon 2B of the two genes.

### 3.2. Clinical Demographic Data

The clinical demographic data showed that the mean age at diagnosis was 50.6 years. Most of the patients were in the age range of 31 to 50 years (34%), although women between 31 and 65 years of age had a similar frequency (30%). Both a basic elementary school education (54%) and low–middle socioeconomic status (60%) were observed in the patients. Further, 64% of the cases had no family history of neoplasms and 96% were HPV-negative. Most of the patients denied the consumption of tobacco or alcohol (84% and 80%, respectively). Finally, 46% of the cases were in FIGO (International Federation of Gynecology and Obstetrics) stage IIB (Table 4).

## 4. Discussion

Currently, despite the large amount of information available and the advances in the practice of oncopathology, the incidence and prevalence of various types of cancer are high and, unfortunately, the diagnosis in most patients is performed at advanced stages, on the one hand, due to the absence of regular medical examinations, and on the other hand, due to the absence of reliable markers that identify specific oncogenic molecules. Therefore, this disease remains the leading cause of death worldwide [19]. Cervical cancer is no exception. Despite the different prevention and management campaigns, this multifactorial neoplasm is one of the most prevalent in the female population (ranked fourth), for which its etiology continues to be studied around the world [32]. Evidence in the literature has established that one of the many factors that could represent a risk for the development of the disease is genetics; therefore, a large number of investigations are directed at analyzing the aberrant expression of several candidate genes [33], as well as searching for gene variants [13], in order to identify targets that allow an effective diagnosis, stratify the cancer into stages, suggest timely treatment, and consequently reduce morbidity and mortality rates [19].

Particularly, *HOX* genes are suggested as biomarkers for the early detection of cancer, first because they are master regulators responsible for controlling tumor initiation and growth, invasion and metastasis, angiogenesis, and resistance to anticancer drugs; therefore, the deregulation of these genes results in the abnormal development and formation of malignant tumors in humans. Furthermore, the study of the regulatory mechanisms of *HOX* genes in tumor development may be key to the diagnosis, treatment, and prognosis of the disease [34]. Second, due to their ability to behave as oncogenic transcriptional factors, regulating multiple pathways that are critical for the malignant progression of a variety of tumors, and third, because the evidence in the literature has reported differences in expression patterns in some types of cancer, and in this context, the abnormal expression of *HOX* genes could affect cell proliferation, differentiation, apoptosis, motility, angiogenesis, autophagy, and cellular receptor signaling [35]. Additionally, it has been documented that certain *HOX* genes are overexpressed, while others show decreased expression, therefore some function as oncogenes (activators) and others as tumor suppressor genes (repressors). In cancer, overexpression is frequent and correlated with more aggressive tumors, leading to an unfavorable prognosis [36]. The differences in gene expression have been attributed to the type, site, and *HOX* gene involved in their development. However, to date, a detailed analysis of each member of the family of these important genes, in which the transformation of normal cells to cancer cells is assessed, has not been performed. Neither are there any studies associating all *HOX* genes with all types of cancer, therefore the findings remain controversial. Under these arguments, it is clear that *HOX* genes continue to attract attention; therefore, some research work is being proposed in different populations around the world [37]. In this regard, the paralogous genes of group 13 (A-D13) play an important role in the development of genitourinary structures, which has made them of particular interest in the study of CC [8,38].

To the best of our knowledge, this is the first description of *HOXC13* and *HOXD13* gene variants in Mexican patients with cervical cancer. Despite the fact that the synonymous variants c.777C>T p.(Arg259Arg), in the *HOXC13* gene, and c.204G>A p.(Ala68Ala) and c.267G>A p.(Ser89Ser), in the *HOXD13* gene, do not result in an amino acid change, previous reports have suggested that, as a result of evolutionary pressure, they may have a potential effect on alternative splicing and translation speed, which could result in a dysfunctional protein associated with diseases [39]. The *HOXC13* gene variant c.895C>A p.(Leu299Ile) is of particular interest due to its location within the homeodomain (exon 2), an important site that enables *HOX* proteins to bind to specific DNA sites to regulate transcriptional activity and to activate or repress target genes [14,34] and because the bioinformatics analysis for this and the *HOXD13* gene variant c.128T>A p.(Phe43Tyr) predicted a deleterious effect on protein structure and function, supporting the idea that these genetic changes could be a risk factor in the development of CC, in addition to the statistical significance (P=0.001) observed for the latter variant (Table 2). Although several bioinformatics servers exist, SIFT and PolyPhen-2 are widely referenced for predicting the impact of non-synonymous single nucleotide variants (NSVs), since they contain evolutionary information and descriptive features of protein structure and function [40,41]. However, PolyPhen-2 has high fidelity in disease association; hence, it is commonly used to corroborate predictions made with other servers [42].

Determining the oncogenic potential of non-synonymous gene variants, in addition to any changes they may have on the coded protein, is as important as it is needed. For such an end, the CGI server was used, yielding the results shown in this study. On one hand, the analysis indicated that both *HOXC13* and *HOXD13* can be held as mutational cancer drivers in breast adenocarcinomas and esophageal tumors, respectively, highlighting their proliferation-inducing capacity. However, it must be mentioned that the role that these genes may have in CC has not been analyzed yet. On the other hand, the non-synonymous variants c.895C>A p.(Leu299Ile) and c.128T>A p.(Phe43Tyr) (*HOXC13* and *HOXD13*, respectively) were classified by the CGI server as Passenger-type mutations, which are typically neutral or have minimal biological consequences, in contrast with Driver-type mutations, although the role and function of the latter remain unclear [43]. Previous studies suggest that Passenger mutations could diminish cell proliferation and could, therefore, be used for therapeutic purposes [44], although other authors have proposed that the accumulation of Passenger mutations could facilitate access to alternative oncogenic pathways, accelerating tumor evolution and thus having a similar effect to Driver mutations. With this in consideration, their classification and biological significance should be more cautious [43]. Likewise, it should be highlighted that the predictions made by the CGI server are also based on previously reported tissue-specific mutations [45], and the hereby shown gene variants have not been previously identified nor linked with cervical cancer; consequently, we consider their potential use as a baseline for future studies, especially because, according to previous reports, the mechanisms and pathways by which *HOX* genes generate cancer are varied and are only beginning to be understood. In some tissues, as mentioned, these genes act as tumor suppressors, and in others, they have been associated with oncogenesis. In this regard, two main mechanisms have been proposed. The first is that *HOX* genes are expressed in a specific tissue, thus tumor formation is temporospatial in relation to normal tissue. The second mechanism suggests gene dominance in which *HOX* genes are expressed at a high level not seen in that tissue type. Additionally, these genes, as mentioned above, could affect various pathways such as the receptor signaling pathway, promoting tumor formation and in some cases metastasis, or the protein–protein interaction pathway, which could lead to the same result [15].

Furthermore, regarding the clinical demographic characteristics, we observed that most of the included patients were HPV-negative (96%), supporting the idea that HPV infection is necessary but not sufficient to result in the development of CC and that the involvement of additional factors (e.g., genetics) could have a synergic effect [46]. In this sense, 36% of them had at least one family member with some type of neoplasm; in addition, the hereby identified non-synonymous variants were found in patients with a family history of cancer. According to previous reports, the susceptibility toward the development of cancer is greater with a positive family history of the disease [47]. Although, in this first study, none of the variants identified in our women with CC were searched for in other first-degree relatives, which would be recommended in subsequent analyses in order to determine the possible involvement of Mendelian inheritance. On the other hand, it has been reported that the Hispanic/Latina population has the highest morbidity and mortality rate of CC, as well as the worst progression compared to the non-Hispanic/Latina population. For this reason, it is essential to analyze the molecular and genetic basis for the development of this neoplasm in different ethnic groups [48,49].

In addition to this, the mean age at diagnosis as well as the age range of our patients was consistent with that reported worldwide (50 years vs. 53 years, respectively, and 31–65 years vs. 35–65 years) [4,10]. The education level (basic elementary) and socioeconomic status (medium-low), determined in more than half of the affected women (54% and 60%, respectively), suggest that healthcare misinformation or non-attendance at health institutions may be key factors in some cases. Tobacco and alcohol consumption were irrelevant in this study (84% and 80%, respectively, denied their use), in contrast to other reports in which these factors were associated with the disease [50]. Finally, several of our patients (46%) were diagnosed at FIGO stage IIB, i.e., with locally advanced cervical cancer, where the estimated 5-year survival rate for these cases is only 40–50%, despite treatment [7]. Therefore, it is important to look for new clinically relevant markers.

New and interesting results were generated in this initial study, although some considerations should be discussed. First, regarding the size of the sample, additional studies in larger patient populations are needed to support the results observed. However, it must be considered that populations may behave differently. In some of them, for example, no gene variant is observed despite the increased sample size, or the association with disease is unclear. Second, it would be interesting to search for and analyze these variants in other ethnic groups in order to determine whether our results are consistent with those obtained in other populations or whether the variants identified are geographically restricted. Finally, in vivo studies should be performed to confirm the functional impact of the proteins predicted in this study.

## 5. Conclusions

Nowadays, molecular techniques offer multiple advantages over other conventional methods for cancer detection. Establishing the precise role of *HOX* genes in neoplastic processes and metastasis would allow their use as molecular markers and therapeutic targets to reduce the number of cases and the mortality of affected individuals.

Differences in relation to genetic susceptibility to cancer in individuals have attracted the attention of researchers and health institutions. To date, the analysis of the *HOXC13* and *HOXD13* genes in cervical cancer is limited to expression studies; thus, the results generated in this initial research work have contributed new information in the hope of enriching the molecular basis of the disease.

Here, we identified five unreported variants in the *HOXC13* and *HOXD13* genes. The non-synonymous variants, c.895C>A p.(Leu299Ile) and c.128T>A p.(Phe43Tyr), could represent a risk factor for the development of cervical cancer. However, additional studies in larger patient populations and in different ethnic groups are needed to support the observed results.

## Figures and Tables

**Figure 1 genes-14-00358-f001:**
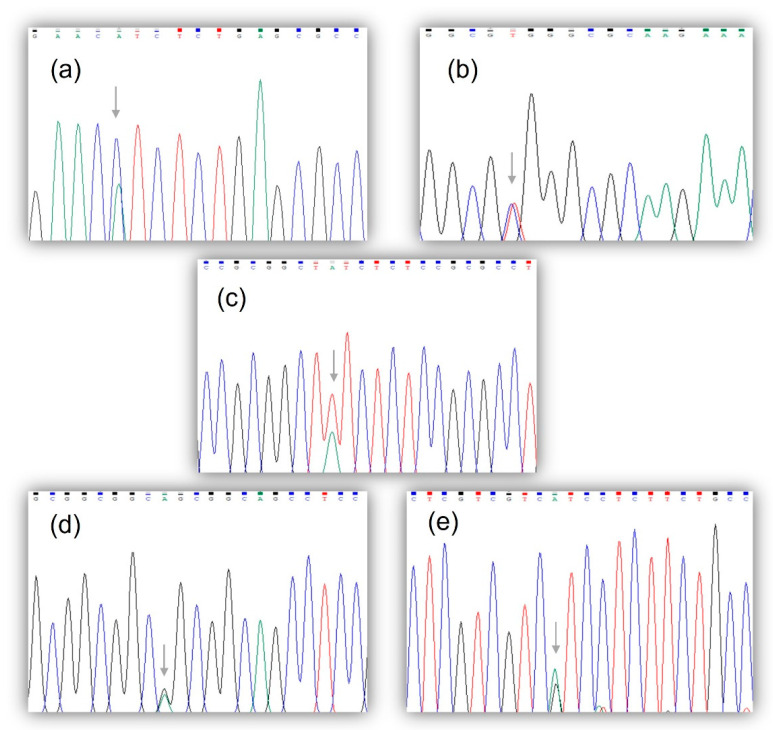
*HOXC13* and *HOXD13* gene variants identified in CC patients. Electropherograms showing the non-synonymous transversion, c.895C>A p.(Leu299Ile), and the synonymous transition, c.777C>T p.(Arg259Arg), both in a heterozygous state (**a**,**b**, respectively), as well as the non-synonymous transversion, c.128T>A p.(Phe43Tyr), and the synonymous transitions, c.204G>A p.(Ala68Ala) and c.267G>A p.(Ser89Ser), all of which were observed in a heterozygous state (**c**–**e**, respectively).

**Table 1 genes-14-00358-t001:** *HOXC13* and *HOXD13* primers.

Gene	Forward	Reverse	Exon	Fragment Size	Conditions
*HOXC13*	5′-GTGTCTCCGC ATGCGTAGAG-3′	5′GGAAGGGAGA CTTCCAGAGG-3′	1	907 pb	95 °C for 5 min, followed by 35 cycles at 95 °C for 1 min; 59, 53, and 54 °C, respectively, for 1 min; 72 °C for 52 s; and 72 °C for 7 min.
	5′-ACTTCTTCCC GCTTGCCTTA-3′	5′-GAAATCTTGC CTAAGGAGTG-3′	2A	789 pb	
	5′-AATTCTTGCC TCATCCTATG-3′	5′-AGTACATTGT CATTCAGACA-3′	2B	848 pb	
*HOXD13*	5′-AGAGAGGGCT AGAGGAAGAG-3′	5′-GGCTGGTCCT TGGTGCAGTA-3′	1	874 pb	95 °C for 5 min, followed by 35 cycles at 95 °C for 1 min; 59, 53, and 53 °C, respectively, for 1 min; 72 °C for 52 s; and 72 °C for 7 min.
	5′-GCTCCGAATA TCCCAGCCTA-3′	5′-GAAGATAATC AGTGCTGGGA-3′	2A	701 pb	
	5′GAAGTGCCAT TCTGATTTAA-3′	5′-AAGAGTTCTG TTTATTGGCA-3′	2B	872 pb	

**Table 2 genes-14-00358-t002:** Allelic and genotypic frequencies of *HOXC13* and *HOXD13* gene variants.

Gene	Variant	Amino Acid Change	Exon	Genotype	CC Patient Frequency (%)	Healthy Women Frequency (%)	*p* Value	Allele	Allelic Frequency (CC Patients %)	Allelic Frequency (Healthy Women %)	*p* Value
*HOXC13*	c.895C>A	p.Leu299Ile	2A	CC	49 (98)	50 (100)	0.315	C A	99 (99)	100 (0)	0.316
				CA	1 (2)	0 (0)			1 (1)	0 (0)	
				AA	0 (0)	0 (0)					
				Total	50 (100)	50 (100)			100 (100)	100(100)	
	c.777C>T	p.Arg259Arg	2A	CC	49 (98)	50 (100)	0.315	C T	99 (99)	100 (0)	0.316
				CT	1 (2)	0 (0)			1 (1)	0 (0)	
				TT	0 (0)	0 (0)					
				Total	50 (100)	50 (100)			100 (100)	100(100)	
*HOXD13*	c.128T>A	p.Phe43Tyr	1	TT	39 (78)	50 (100)	0.000 *	T A	89 (89)	100 (0)	0.001 *
				TA	11 (22)	0 (0)			11 (11)	0 (0)	
				AA	0 (0)	0 (0)					
				Total	50 (100)	50 (100)			100 (100)	100(100)	
	c.204G>A	p.Ala68Ala	1	GG	48 (96)	46 (92)	0.360	G A	98 (98)	94 (94)	0.149
				GA	2 (4)	2 (4)			2 (2)	6 (6)	
				AA	0 (0)	2 (4)					
				Total	50 (100)	50 (100)			100 (100)	100(100)	
	c.267G>A	p.Ser89Ser	1	GG	49 (98)	47 (94)	0.360	G A	99 (99)	95 (95)	0.097
				GA	1 (2)	1 (2)			1 (1)	5 (5)	
				AA	0 (0)	2 (4)					
				Total	50 (100)	50 (100)			100 (100)	100(100)	

* *p* < 0.05.

**Table 3 genes-14-00358-t003:** Bioinformatic analysis of coding variants in *HOXC13* and *HOXD13* genes.

Gene	Exon	Variant	Bioinformatics Server	Score	Effect	Interpretation
*HOXC13*	2A	c.895 C>A (p.Leu299Ile)	SIFT	0	Damaging	Amino acid substitution is deleterious. Affects protein function.
			PolyPhen-2	1	Probably Damaging	Amino acid substitution is damaging. Affects protein structure and function.
*HOXD13*	1	c.128T>A (p.Phe43Tyr)	SIFT	0.002	Damaging	Amino acid substitution is deleterious. Affects protein function.
			PolyPhen-2	0.898	Possibly Damaging	Amino acid substitution is damaging. Possibly affects protein structure and function.

**Table 4 genes-14-00358-t004:** Clinical demographic characteristics of the cervical cancer patients included in this study *(n =* 50).

Characteristic	N° of Cases	%
Age at diagnosis		
≤30 years	8	16
31–50 years	17	34
51–65 years	15	30
>65 years	10	20
Family history		
Yes	18	36
No	32	64
Schooling		
None	16	32
Elementary	27	54
High school	4	8
Bachelor degree	3	6
Socioeconomic status		
Low	13	26
Lower-middle	30	60
High	7	14
HPV		
Yes	2	4
No	48	96
Alcohol		
Yes	10	20
No	40	80
Tobacco		
Yes	8	16
No	42	84
FIGO system		
I	10	20
II	23	46
III	6	12
IV	11	22

## Data Availability

All data used in this paper are available in the article.

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
