# Peer review of "Variants Identified in the HOXC13 and HOXD13 Genes Suggest Association with Cervical Cancer in a Cohort of Mexican Women"

_genes, 2023, doi:10.3390/genes14020358_

Round 1
Reviewer 1 Report
In this manuscript, the authors investigated the HOXC13 and HOXD13 gene variants in Mexican females with cervical cancer. They identified five novel variants, among which 2 are non-synonymous variants. The frequency of the presence of c.128T>Ap. variant was significant between patients and the control group. The authors also summarized the relevant demographic information of the cc patients. Here are my comments:
1. In figure 1, the authors should include longer flanked genomic sequencing results rather than just 2-3 bp. It would be appreciated if they could indicate the location of each mutation in the HOXC13 and HOXD13 genes.
2. Are the two gene loci that have non-synonymous mutation conservative across species? The authors should include this information in the manuscript.
3. The authors mentioned that synonymous mutations may cause alternative splicing. Is possible to extract mRNA from peripheral blood and test that hypothesis?
4. The authors should also include the demographic information such as schooling, socioeconomic, etc. of the control group for comparison
Author Response
Dear reviewer, we appreciate your requests, we responded one by one which you can see in the attached file.
Thanks for your time.

Reviewer 2 Report
Good study
Author Response
Does not apply.
Reviewer 3 Report
In this study, the authors performed PCR on 50 Mexican women with cervical cancer as well as 50 healthy women, identified five unreported variants in the HOXC13 and HOXD13 genes. Two nonsynonymous variants (c.895C>A p.(Leu299Ile) and c.128T>A p.(Phe43Tyr) might represent a risk factor for the development CC. More data should be provided to have a solid conclusion.
1. For all the description about “p.(Leu299Iso)” should be corrected as “p.(Leu299Ile)”, and the Nucleotide ID # for both HOXC13 and HOCD13 should be addressed in the molecular analysis part.
2. It would be better if the authors can add some more information regarding cervical cancer, such as the incidence frequency for introduction part. Also, the authors should clarify the clinical phenotype inclusion criteria of Mexican women with cervical cancer in the methods, which will give more clarification about the case selection criteria.
3.The variants in HOXC13 was reported to cause ectodermal dysplasia (OMIM #142976), and variants in HOXD13 was proved to cause syndactyly syndrome (OMIM #142989), raising the question: Besides Cervical Cancer, do individuals carrying variants in HOXC13 and HOXD13 gene share other clinical features?
4. The authors should add more details of candidate variants screening criteria (inclusion/exclusion) properly in the method part, including the inheritance pattern of autosomal dominant or recessive, as well as the variants frequency filter screening criteria.
5. Because the altered expression of HOXC13 and HOXD14 has been associated to several types of cancers, the expression level of identified variants in HOXC13/HOXD14 should be further explored by quantitative Real-time PCR as well as western blot to detect both mRNA level and protein expression level, which could offer much more solid evidence to establish the causality between the variants and the disease.
6. Even the allelic frequency of variants was significantly different between patients and health controls (50/50, limited number of cohort), the variants should be further evaluated in a larger database such as genome Aggregation Database (gnomAD) or similar one, especially for the non-Latino-American ethnicity.
7. Since the identified variants has no experimental evidence for supporting the pathogenic effect, the title “Pathogenic Variants in HOXC13 and HOXD13 Genes of Mexi-can Women with Cervical Cancer” should be adjust into variants association with Women with Cervical Cancer instead of pathogenic cause.
Author Response
Dear reviewer,
On behalf of all co-authors, we appreciate your requests, we responded one by one, which you can see in the attached file.
Thanks for your time.

Round 2
Reviewer 3 Report
Most of my remarks have been answered adequately. However, further attention should be paid to:
1. For the discussion part, the limitations of this study should be discussed deeply and properly, which including the limited number of cases, multiple ethnics could be explored, potential biological supportive evidence which could be further studied in vitro and in vivo, et. al.
2. Here is another question: did the author ever confirm the variants from the gnomic DNA of patients’ father, mother, or siblings? It would be more helpful and instructive to provide the Mendelian inheritance evidence for further clinical genetic screening. Or else, the related discussion should be addressed properly in the Discussion Part.
Author Response
On behalf of all the authors of the work, we appreciate the comments, observations, and requests. Please see the attachment. The most recent changes are marked on lines 183-186 and 195-204.
|
Suggested comment. |
Corrections or comments. |
|
1. For the discussion part, the limitations of this study should be discussed deeply and properly, which include the limited number of cases, multiple ethnics that could be explored, potential biological supportive evidence which could be further studied in vitro and in vivo, et. al. |
Change made: The discussion was modified. |
|
2. Here is another question: did the author ever confirm the variants from the genomic DNA of the patients’ fathers, mothers, or siblings? It would be more helpful and instructive to provide Mendelian inheritance evidence for further clinical genetic screening. Or else, the related discussion should be addressed properly in the Discussion Part. |
Change made: The discussion was modified.
|
Thank you.
